# Person-Centred Care: A Support Strategy for Managing Non-Communicable Diseases

**DOI:** 10.3390/healthcare12050526

**Published:** 2024-02-23

**Authors:** Mateja Lorber, Nataša Mlinar Reljić, Barbara Kegl, Zvonka Fekonja, Gregor Štiglic, Adam Davey, Sergej Kmetec

**Affiliations:** 1Faculty of Health Sciences, University of Maribor, 2000 Maribor, Slovenia; natasa.mlinar@um.si (N.M.R.); barbara.kegl@um.si (B.K.); zvonka.fekonja@um.si (Z.F.); gregor.stiglic@um.si (G.Š.); sergej.kmetec1@um.si (S.K.); 2College of Health Sciences, University of Delaware, Newark, DE 19716, USA; davey@udel.edu; 3Permanent Working Group of Palliative Care, Nurses and Midwives Association of Slovenia, 1000 Ljubljana, Slovenia

**Keywords:** person-centred care, non-communicable disease, quality of life, life satisfaction

## Abstract

Background: Over the last decade, the inadequacy and unsustainability of current healthcare services for managing long-term co-morbid and multi-morbid diseases have become evident. Methods: This study, involving 426 adults with at least one non-communicable disease in Slovenia, aimed to explore the link between quality of life, life satisfaction, person-centred care, and non-communicable disease management. Results: Results indicated generally positive perceptions of quality of life, general health, and life satisfaction of individuals with non-communicable diseases. Participants assessed their physical health as the highest of the four quality of life domains, followed by the environment, social relations, and psychological health. Significant differences occurred in life satisfaction, general health, quality of life, and person-centred care for managing non-communicable diseases. But, there were no significant differences in person-centred care according to the living environment. The study revealed a positive association between person-centred care and effective non-communicable disease management, which is also positively associated with quality of life, general health, and life satisfaction. Conclusions: Person-centred care is currently the most compassionate and scientific practice conceived, representing a high ethical standard. However, implementing this approach in healthcare systems requires a cohesive national strategy led by capable individuals to foster stakeholder collaboration. Such an approach is crucial to address the deficiencies of existing healthcare services and ensure person-centred care sustainability in non-communicable disease management.

## 1. Introduction

Also referred to as chronic diseases, non-communicable diseases (e.g., cardiovascular diseases, cancers, chronic respiratory diseases, diabetes and others) typically persist for extended periods and arise from genetic, physiological, environmental, and behavioural influences. Annually, non-communicable diseases are responsible for 41 million deaths, which accounts for 74% of global fatalities, according to the World Health Organization [1]. The prevalence of non-communicable diseases escalates with age. Although the discourse on non-communicable diseases is predominantly centred on prevention, control, and the socio-economic factors influencing health, it is crucial to note that the overall impact of non-communicable diseases is also shaped by the demographic composition and age distribution of a country’s population [2]. Non-communicable diseases present severe health risks for individuals, families, and communities and pose a potential threat to the capacity of healthcare systems. Given the significant socio-economic costs of non-communicable diseases, their prevention and control have become a crucial development goal for the 21st century [3]. It is also important to note that the quality of extended life—whether in good health or with illness—is vital for health system planners. They must comprehend the healthcare ramifications of an ageing population that could potentially experience long-term illnesses and complex multimorbidity [4]. Age is the primary determinant of risk for non-communicable diseases, as evidenced by studies conducted by Niccoli and Partridge [5] and Hou et al. [6]. However, the extent of age-related decline varies significantly among individuals, as noted by Elliott et al. [7] and Tuttle et al. [8]. It is noteworthy that a substantial proportion of the adult population, approximately half, grapples with one or more non-communicable diseases [5].

Consequently, individuals facing such conditions must confront many day-to-day challenges encompassing diverse symptoms, complications, loss of independence, and the self-management of non-communicable diseases. Additionally, it is pertinent to highlight that psychological distress experienced in the context of non-communicable diseases is linked to a decline in self-assessed health [9], a diminution in disease self-management, and adherence to lifestyle recommendations [10,11]. On a global scale, non-communicable diseases significantly impact numerous individuals’ overall well-being and quality of life, exacerbated by rapid and unstructured urban development, the widespread adoption of detrimental habits, and an ageing population [12].

Distressing symptoms, such as fatigue, difficulties with sleep, weakness, and social seclusion, among others that arise from non-communicable illnesses, significantly diminish the well-being of individuals [13,14]. Individuals suffering from non-communicable diseases are frequently left to their own devices, necessitating substantial effort and time from themselves and their family members to make disease management decisions [15,16]. The provision of individualised care assumes a critical role in managing non-communicable diseases [17].

Person-centred care embodies a comprehensive and integrated methodology that esteems each individual as a distinct entity, encompassing their unique needs and values. This strategic approach customises care to address the specific non-communicable disease and the individual’s condition, offering comprehensive care that attends to their physical, psychological, spiritual, and social requirements [17,18]. There is a lot of evidence that person-centred care is an effective therapeutic intervention for different patient outcomes [19,20,21,22,23]. In their rapid literature review, Cano et al. [22] discovered that implementing per-son-centred care models to foster self-care empowerment in long-term care resulted in multidimensional health-related outcomes. These outcomes have implications not only at the individual level but also at institutional and societal levels [22]. The development of self-awareness and the motivation for change can be fostered through personality assessments that acknowledge the intricate psychobiological makeup of human personality. The process of personality transformation is contingent on the interplay between an individual’s physical, mental, and spiritual facets [23]. Person-centred care enhances individuals’ quality of life and satisfaction and alleviates emotional strain on family members.

Moreover, this approach mitigates depression [24]. The primary objective of person-centred care is to uphold and enhance the quality of life for patients, their family members, and caregivers. Cultivating a sense of shared responsibility and collaborative decision-making in providing this type of care is crucial. A comprehensive body of literature, comprising 23,000 studies, has investigated the concept of person-centred care, either as a primary focus or as an integral component within broader research inquiries. Within this extensive collection, a meticulous review has discerned 921 studies specifically dedicated to exploring person-centred care. Among these, 503 studies directly employed methodologies tailored to measure the application of person-centred care, while 418 studies delved into associated aspects of this care paradigm. Methodological frameworks used within these studies encompassed a variety of approaches, including cross-sectional analyses (59%), qualitative methods such as interviews or focus groups (10%), observational techniques (6%), multi-methodological approaches (5%), and other diverse methodologies (8%) (de Silva, 2014). Furthermore, these investigations were geographically widespread, spanning regions such as the United Kingdom, Europe, North America, and other global territories [25].

Additionally, person-centred care is correlated with an improved quality of life and heightened satisfaction with care [26,27]. The person-centred approach underscores the humanisation of healthcare [26,28]. Berghout et al. [29] discovered that healthcare professionals also report positive experiences when delivering person-centred care.

Healthcare professionals’ inadequate knowledge often impedes the implementation of person-centred care in the healthcare sector, creates a disregard for the person-centred approach, and highlights their inability to adapt to the individual’s health condition. These obstacles can decrease individuals’ quality of life and satisfaction, potentially leading to social isolation, despair, and self-identity loss [30,31].

Until now, the analysis of existing studies indicates a scarcity of research encompassing satisfaction with life, overall health, and the practice of person-centred care among individuals with non-communicable diseases. As a result, we aim to investigate the association between general health, life satisfaction, person-centred care, and the management of non-communicable diseases among adults with at least one non-communicable disease.

## 2. Materials and Methods

### 2.1. Design

The study used a quantitative cross-sectional methodology, using a validated survey instrument to gather data on various aspects, including overall health, life satisfaction, person-centred care, and disease management from individuals with non-communicable diseases. The study results’ credibility was assured through strict adherence to the STROBE (Strengthening the Reporting of Observational Studies in Epidemiology) checklist [32].

### 2.2. Setting and Sample

This study used convenience sampling to recruit adult participants from the northeastern region of Slovenia. The sample selection was based on inclusion criteria: adult participants with at least one non-communicable disease. Conversely, the exclusion criteria encompassed participants younger than 18 (not adults) and those not diagnosed with a non-communicable disease. Following the Cochran formula [33] and considering the number of adults suffering from chronic disease (*n* = 433.230, recognising that this number is subject to constant fluctuations due to newly diagnosed diseases), our estimation revealed that the necessary sample size should consist of 384 participants. Out of the 640 questionnaires that were distributed, 439 were returned, resulting in a response rate of 69%. Following this, 13 questionnaires were excluded due to a non-completion rate of 50%, ultimately leading to a final sample size of 426 participants.

### 2.3. Study Tools/Instruments

Participants in this study completed a self-report questionnaire, including the World Health Organization Quality of Life-Bref version (WHOQOL-BREF) [34], the Satisfaction with Life Scale (SWLS) [35], and the Person-centred Practice Inventory for Service Users (PCPI-SU) [36], as well as questions about their demographics, such as physical activity, managing a non-communicable disease, income, social interaction, and support of loved ones.

The assessment of participants’ quality of life was completed by utilising the WHOQOL-BREF [34], a questionnaire consisting of 26 items. This instrument encompasses four domains: physical health (7 items), psychological health (6 items), social relationships (3 items), and environment (8 items). Additionally, supplementary items are included to measure general health (1 item) and overall quality of life (1 item). Each item was evaluated by respondents using a 5-point Likert scale, where higher scores indicate an improved quality of life. The scores for each domain were calculated by summing the item scores within that specific domain and subsequently transformed to a 0–100 scale linearly. A score of 0 signifies the lowest possible health state, while a score of 100 indicates the highest possible health state within the respective domain.

Participants answered the question about managing non-communicable diseases on a four-point scale (very good to very poor).

The SWLS [35] questionnaire evaluated the participants’ life satisfaction and consisted of five items. Respondents expressed their agreement or disagreement with each item using a 7-point scale, ranging from 7 (strongly agree) to 1 (strongly disagree). Possible scores range from 5 to 35, with predefined cut-offs categorising satisfaction levels as Extremely satisfied (31–35), Satisfied (26–30), Slightly satisfied (21–25), Neutral (20), Slightly dissatisfied (15–19), Dissatisfied (10–14), and Extremely dissatisfied (5–9).

The PCPI-SU [36], a 20-item questionnaire, aimed to capture individuals’ perceptions of preserving person-centred care. This instrument encompasses five domains: working with the Person’s Believe and Values (4 items), Sharing Decision-making (5 items), Engaging Authentically (4 items), Being Sympathetically Present (3 items) and Working Holistically (3 items). Grounded in McCormack and McCance’s concepts of person-centred care [28], each statement was rated on a 5-point Likert scale, spanning from ‘strongly disagree’ (1) to ‘strongly agree’ (5). Higher scores on this instrument denoted a more positive perception of preserving person-centred care.

### 2.4. Data Collection

Data were gathered between January and June 2023, with previous ethical approval. The research team personally distributed questionnaires to all eligible participants and explained the study’s aims to them. The participants were requested to complete the instrument conveniently and return it to the researcher in a sealed envelope within seven days at an agreed place. Following data collection, the information was entered into the SPSS Statistic 25 software (SPSS Inc., Chicago, IL, USA).

### 2.5. Data Analysis

This study’s statistical analysis used both descriptive and inferential statistics. For demographic data, descriptive statistics, such as frequency, percentages, and mean and standard deviation estimates, were used. Categorical variables were presented as frequencies and percentages. The Shapiro–Wilk test was utilised to assess the normality distribution of variables. The results of this test indicated a departure from the normal distribution of the data. Considering the non-normal distribution, differences between the groups were assessed using either the Kruskal–Wallis or Mann–Whitney U-test. The Spearman correlation coefficient explored the associations among general health, life satisfaction, person-centred care, and non-communicable disease management. The significance level for all analyses was set at less than 0.05 to establish statistical significance. Furthermore, the internal consistency and reliability of the WHOQOL-BREF [34], SWLS [35], and PCPI-SU [36] questionnaires were evaluated using Cronbach’s α coefficient.

### 2.6. Ethical Considerations

Before conducting the study, formal approval was sought from the relevant ethical committee (Ref. No: 038/2022/5006-4/902; approval 29 September 2022). Additionally, we obtained permission to conduct the study from nursing homes, health centres, and individual participants. Stringent ethical protocols were followed, with participants being presented with a participant information sheet attached to the first page of the questionnaire, delineating the study’s aim and providing all necessary information. Subsequently, written consent was obtained from each participant. In recognition of the non-invasive nature of the study and the absence of any risks or hazards to the participants, informed verbal consent was deemed appropriate. The ethical conduct of the study adheres strictly to the principles outlined in the Declaration of Helsinki [37] and incorporates the provisions of the Oviedo Convention [38]. This ethical framework ensures the protection of participants’ rights and upholds the integrity of the research process.

## 3. Results

The survey involved 426 participants from the northeastern region of Slovenia. The calculated Cronbach’s α coefficients were 0.897 for WHOQOL-BREF, 0.876 for SWLS, and 0.816 for PCPI-SU. These coefficients reflect a high internal consistency and reliability level, affirming that the instruments consistently measure the intended constructs accurately and precisely. The demographics of the participants are presented in Table 1.

A total of 426 patients with at least one non-communicable disease participated in the study. Among them, 65% were female, while 35% were male. The participants comprised 20% of individuals residing in nursing homes and 80% of community-dwelling adults with at least one non-communicable disease. The average age of the participants was 56.32 years (*SD* = 17.57). A substantial portion (42%) were married, and 25% cohabited. More than half (52%) of the participants had attained at least a secondary level of education, and a similar proportion (58%) reported having one non-communicable disease (Table 1).

From Table 2 we can see that according to the WHO-QOL-BREF, participants assessed as the highest physical health (*M* = 65.31; *SD* = 14.9; 95%; IC = 63.89–66.74), followed by the environment (*M* = 62.99; *SD* = 15.1; 95%; IC = 61.15–64.41); social relations (*M* = 59.49; *SD* = 19.5; 95%; IC = 57.71–61.25) and psychological health (*M* = 54.37; *SD* = 10.0; 95%; IC = 53.45–55.21). The highest value for person-centred care was the person’s beliefs and values (*M* = 3.41; *SD* = 0.93; 95%; IC = 3.32–3.50), followed by the role of working holistically (*M* = 3.31; *SD* = 0.86; 95%; IC = 3.22–3.39), being sympathetically present (*M* = 3.30; *SD* = 0.87; 95%; IC = 3.22–3.38), sharing decision-making (*M* = 3.30; *SD* = 0.89; 95%; IC = 3.21–3.39) and engaging authentically (*M* = 3.29; *SD* = 0.86; 95%; IC = 3.21–3.37). The participant’s quality of life score was 3.77 (*SD* = 0.72; 95%; IC = 3.70–3.85) and general health was 3.40 (*SD* = 1.01; 95%; IC = 3.31–3.50). Participants’ life satisfaction averaged 24.23 (*SD* = 5.9; 95%; IC = 23.67–24.80).

As shown in Table 3, data indicate a significant difference in life satisfaction (*H*(4) = 26.836; *p* < 0.001), general health (*H*(4) = 53.343; *p* < 0.001), quality of life (*H*(4) = 38.717; *p* < 0.001) and person-centred care (*H*(4) = 19.330; *p* < 0.001) according to relationship status. There is a significant difference in life satisfaction (*H*(3) = 22.590; *p* < 0.001), general health (*H*(3) = 23.174; *p* < 0.001) and quality of life (*H*(3) = 36.689; *p* < 0.001) according to educational level. A significant difference was found in the quality of life according to the number of non-communicable diseases (*H*(1) = 10.431; *p* < 0.001) and frequency of physical activity (*H*(4) = 27.186; *p* < 0.001). Significant differences were found in life satisfaction (*H*(3) = 18.128; *p* < 0.001), general health (*H*(3) = 45.364; *p* < 0.001), quality of life (*H*(3) = 33.386; *p* < 0.001) and person-centred care (*H*(3) = 11.622; *p* = 0.009) according to the managing non-communicable disease.

We also checked if any differences exist in the four domains of quality-of-life assessment. There were no differences according to gender, but there were significant differences in physical health (*H*(4) = 15.909; *p* < 0.001), psychological health (*H*(4) = 23.420; *p* < 0.001), social relations (*H*(4) = 34.858; *p* < 0.001) and environment (*H*(4) = 25.124; *p* < 0.001) according to relationship status. According to the number of non-communicable diseases, there were differences only in physical health (*H*(1) = 7.410; *p* = 0.006). The differences in physical health (*H*(3) = 12.558, *p* = 0.006), social relations (*H*(3) = 16.317, *p* < 0.001) and environment (*H*(3) = 13.439, *p* = 0.004) were found according to the managing non-communicable disease. Differences were found also in physical health (*H*(4) = 47.395, *p* < 0.001), psychological health (*H*(4) = 17.451, *p* = 0.002), social relations (*H*(4) = 20.023, *p* < 0.001), and environment (*H*(4) = 36.920, *p* < 0.001) according to the frequency of physical activity. 

With Spearman correlation analysis it was found that the better management of non-communicable diseases is associated with physical health (*r_s_* = 0.118; *p* = 0.015), psychological health (*r_s_* = 0.257; *p* < 0.001), social relations (*r_s_* = 0.302, *p* < 0.001), environment (*r_s_* = 0.292; *p* < 0.001), contacts with loved ones (*r_s_* = 0.512; *p* < 0.001), physical activity (*r_s_* = 0.267; *p* < 0.001), economic status (*r_s_* = 0.133; *p* = 0.005), and person-centred care (*r_s_* = 0.129; *p* = 0.009). From the data, we also found that the level of managing non-communicable chronic disease is positively associated with quality of life (*r_s_* = 0.309; *p* < 0.001) and general health (*r_s_* = 0.288; *p* < 0.001). At the same time, quality of life (*r_s_* = 0.589; *p* < 0.001) and general health (*r_s_* = 0.423; *p* < 0.001) are positively associated with life satisfaction.

## 4. Discussion

Our study aimed to evaluate the management of non-communicable diseases, person-centred care, and their association with the quality of life and life satisfaction among adults afflicted with at least one non-communicable disease in Slovenia. The findings underscore the significance of person-centred care as an important variable in effectively managing non-communicable diseases and its influence on life satisfaction and overall quality of life. The data revealed significant differences across various domains, including life satisfaction, general health, quality of life, and person-centred care based on participants’ relationship status. These findings suggest that individuals’ relationship status is pivotal in determining their overall well-being and perception of care. Specifically, those in different relationship statuses may experience varying levels of satisfaction and health outcomes, highlighting the importance of considering relationship dynamics in healthcare interventions to enhance well-being.

Similarly, significant differences in life satisfaction, general health, and quality of life were observed based on participants’ educational levels. This underscores the influence of educational attainment on individuals’ perceptions of health and overall quality of life. Higher levels of education may be associated with better health outcomes and greater life satisfaction, possibly due to increased access to resources, knowledge, and opportunities for self-improvement. The study found significant differences in quality of life based on the number of non-communicable diseases and frequency of physical activity. This highlights the importance of disease management and regular physical activity in enhancing individuals’ quality of life.

Moreover, it suggests that interventions targeting disease prevention and promoting physical activity could improve overall well-being among individuals. Significant differences were observed in various domains, including life satisfaction, general health, quality of life and person-centred care, based on how participants managed non-communicable diseases. This emphasises the need for personalised care approaches tailored to individuals’ disease management strategies, as different approaches may impact their overall well-being and satisfaction with care differently. Examining the quality-of-life domains revealed further insights. While no differences were found according to gender, significant differences were observed in physical health, psychological health, social relations, and environment based on relationship status, number of non-communicable diseases, managing non-communicable diseases, and frequency of physical activity. These findings underscore the multidimensional nature of quality of life and highlight the importance of considering various factors when assessing individuals’ well-being. Moreover, a weak, but consistent, magnitude-positive correlation was observed between enhanced non-communicable disease management and elevated quality of life, general health, life satisfaction, and favourable assessments in physical health, psychological health, social relations, environment, and person-centred care domains. These findings collectively contribute to a more comprehensive understanding of the intricate interplay between disease management, person-centred care, and individuals’ health contending with non-communicable disease in the Slovenian context.

Most studies about person-centred care now focus on hospitalised patients [19,20,39] and older adults in long-term settings [21,22,23,40]. Our research included individuals with at least one non-communicable disease, regardless of the living environment, and we found that physical health, psychological health, social relationships, environment, contact with loved ones, and physical activities were positively associated with individuals’ management of non-communicable disease. There were no significant differences in dimensions of person-centred care, not even in total, between participants according to the living environment, which is encouraging. Those living in the home environment and those living in nursing homes receive the same quality of person-centred care during the treatment of non-communicable diseases.

In the research, participants assessed their physical health as the highest of the four domains when they self-assessed their quality of life, followed by environment, social relations, and psychological health. In the psychological health domain, they barely achieved half the total value (*M* = 54.4; *SD* = 10.0). Compared with the research [41], which included 27 disease groups or health conditions and healthy people from 38 UK sites, they found that the environment was the highest assessed domain, followed by social relations and physical and psychological health. In that research, psychological health was assessed with a higher value (*M* = 62.6; *SD* = 18) than in our research. As in our study, Tseng et al. [41] also discovered a positive correlation between self-care and implementing person-centred care regarding psychological health. This suggests that introducing person-centred care models to encourage self-care empowerment achieves its goals. Tseng et al. [41] also highlighted that research has shown a direct link between general and self-rated health at the individual level.

Individuals already suffering from non-communicable diseases need to comprehend their identity, which enhances their understanding of how their lifestyle impacts their health, quality of life, and life satisfaction. Overall results for quality of life (*M* = 3.77; *SD* = 0.7), general health (*M* = 3.40; *SD* = 1.0), and life satisfaction (*M* = 24.23; *SD* = 5.9) were good. We found that 52% were very satisfied with their health, and 74% assessed their quality of life as good or very good. Our results are better compared to the results of Skevington et al. [42], who found that the quality of life and general health of participants in the UK was good and that 47% of the participants reported their quality of life as good and 37% were satisfied with their health. 

Our research found that person-centred care is positively associated with the better management of non-communicable diseases and that managing non-communicable diseases is positively related to quality of life, general health, and life satisfaction. Other studies [19,20,21,22,23] showing that person-centred care is an effective therapeutic intervention for different patient outcomes support our results. In their rapid literature review, Cano et al. [22] discovered that implementing person-centred care models to foster self-care empowerment in long-term care resulted in multidimensional health-related outcomes with individual, institutional and societal implications.

The domains of person-centred care [43] typically first emerge in midlife; these elements are specifically for older adults with chronic conditions or functional impairments and support our results for adults with non-communicable diseases. According to our results, we should not overlook that it is next to person-centred care, and we have to point out the importance of the environment, social relations, and contacts with loved ones for the better management of non-communicable diseases and consequently better quality of life. We know that non-communicable disease necessitates extended supervision, monitoring, or care. To effectively address non-communicable diseases, healthcare employees must consider the individual’s personal, family, social, political, cultural, and spiritual circumstances. Kogan et al. [43] noted that person-centred care is gaining prominence in all facets of healthcare, particularly in long-term residential care. This is especially true in light of the cultural evolution in caregiving for individuals with significant cognitive and other impairments, which includes various healthcare settings, long-term care, and social and community-based services for older adults.

Several limitations in this study warrant attention. Firstly, the study employed a cross-sectional design, which precludes the ability to draw causal conclusions. There is a scarcity of published studies comparing person-centred care and quality of life, mostly focusing on hospitalised patients and older adults in long-term settings. Secondly, a cross-sectional study can be subject to recall bias. Another potential limitation could be the sampling method used and its impact on the results. Given the predominance of one gender and location in the sample, the results cannot be generalised to the entire population.

Furthermore, the study relied on self-reported data, which could lead to social desirability bias, where respondents might overstate or understate their responses based on societal norms. No significant system failures were identified. The voluntary nature of participation in the study raises questions about whether the sample accurately represents the key characteristics of the population. Lastly, we could not account for all potential confounding factors that could influence the management of non-communicable disease and quality of life.

Despite the limitations, this study represents the initial strategies of healthcare for the improvement of management of non-communicable diseases, which consequently contribute to better quality of life and life satisfaction. This is particularly crucial because the number of older adults and the proportion of individuals with non-communicable diseases is rising. This presents a vital concern that, if not adequately assessed and managed by the healthcare system, it could adversely impact an individual’s ability to fulfil their needs due to a decline in quality of life and life satisfaction. The primary objective of public health is to enhance quality of life, not just to promote independence in daily life. The help of community nurses largely addresses this goal. The necessity for person-centred strategies is evident in the traits of complex adaptive systems, which possess unique properties commonly found in numerous medical conditions. Within these complex adaptive systems, individuals with identical clinical characteristics can experience vastly different results, and conversely, those with varying clinical features may end up with the same outcome.

## 5. Conclusions

The growing trend of the life expectancy of the population, coupled with the prevalence of multiple non-communicable diseases and the heightened susceptibility of older adults, undeniably necessitates person-centred care as an essential strategy. This strategy emphasises the individual’s active involvement in their healthcare treatment, encourages collaborative decision-making and mutual comprehension, and respects their values, preferences, and beliefs. Person-centred care diverts attention from the conventional biomedical approach, instead promoting the personal choice and independence of those who use health services. This strategy has proven to be an essential avenue for improving primary care, with older adults being key focuses of person-centred practice because they are more likely to have their care needs met compared to younger individuals.

## Figures and Tables

**Table 1 healthcare-12-00526-t001:** Participant characteristics.

Variables	Descriptive Statistics Total (*n* = 426)
Gender %(*n*)	
Male	34.7 (148)
Female	65.3 (278)
Age (Year; M ± SD)	56.3 ± 17.6
Relationship Status %(*n*)	
Single	14.6 (62)
Married	41.8 (178)
Divorced	7 (30)
Cohabitation	24.6 (105)
Widowed	12 (51)
Education %(*n*)	
Elementary education	9.2 (39)
Secondary Education	51.9 (221)
HE (Bachelor)	30 (128)
HE (Master or Doctoral)	8.9 (38)
No. of NCD %(*n*)	
One	57.7 (246)
Two to three	40.1 (171)
Four or more	2.1 (09)
Managing NCD %(*n*)	
Very poor	7.5 (32)
Poor	9.2 (39)
Good	75.4 (321)
Very good	8.0 (34)

Note: n—Sample size; %—Percent of participants; M—Mean; SD—Standard deviation; HE—Higher Education; NCD—non-communicable disease.

**Table 2 healthcare-12-00526-t002:** Descriptive data for general health, quality of life, life satisfaction, managing non-communicable disease and person-centred care.

Variables	M ± SD	Min–Max	Me ± IQR
General Health	3.40 ± 1.01	1–5	4.0 ± 1.0
Quality of life	3.77 ± 0.72	1–5	4.0 ± 1.0
Life satisfaction	24.23 ± 5.9	5–35	25.0 ± 9.0
Person-centred care	3.32 ± 0.83	1–5	3.18 ± 1.0
Person’s believe and values	3.41 ± 0.93	1–5	3.25 ± 1.0
Sharing decision-making	3.30 ± 0.89	1–5	3.0 ± 1.0
Engaging authentically	3.29 ± 0.86	1–5	3.0 ± 1.0
Being sympathetically present	3.30 ± 0.87	1–5	3.3 ± 1.0
Working holistically	3.31 ± 0.86	1–5	3.25 ± 1.0
WHOQOL-BREF	60.54 ± 14.87	16.25–95.81	59.88 ± 17.5
Physical health	65.31 ± 14.9	10.75–100	64.5 ± 18.0
Psychological health	54.37 ± 10.0	12.5–83.25	54.0 ± 8.0
Social relationships	59.49 ± 19.5	16.75–100	58.5 ± 25.0
Environment	62.99 ± 15.1	25–100	62.5 ± 19.0
Managing NCD’s	2.16 ± 0.67	1–4	2.0 ± 0

Note: M—Mean; SD—Standard deviation; Me—Median; IQR—Interquartile range; NCD—non-communicable disease.

**Table 3 healthcare-12-00526-t003:** Demographic variables and their relationship with life satisfaction, satisfaction with health, quality of life and person-centred care.

Variables	Total (*n* = 426)	Life Satisfaction M ± SD (Me ± IQR)	General Health M ± SD (Me ± IQR)	Quality of Life M ± SD (Me ± IQR)	Person-Centred Care M ± SD (Me ± IQR)
Gender % (*n*)	—	*U* = 18,861.5 *p* = 0.156	*U* = 19,940.0 *p* = 0.582	*U* = 19,455.0 *p =* 0.277	*U* = 19,183.5 *p =* 0.248
Male	34.7 (148)	24.86 ± 5.49 (27.0 ± 7.0)	3.44 ± 0.97 (4.0 ± 1.0)	3.83 ± 0.69 (4.0 ± 1.0)	3.40 ± 0.81 (3.35 ± 1.0)
Female	65.3 (278)	23.89 ± 6.08 (24 ± 9.0)	3.38 ± 1.04 (4.0 ± 1.0)	3.74 ± 0.74 (4.0 ± 1.0)	3.29 ± 0.83 (3.11 ± 0.96)
Age (year) (*M* ± *SD*)	56.3 ± 17.6	*r_s_ =* −0.243 * *p* < 0.001	*r_s_* = −0.331 * *p* < 0.001	*r_s_* = −0.350 * *p* < 0.001	*r_s_* = 0.199 * *p* < 0.001
Relationship Status % (*n*)	—	*H*(4) = 26.836 * *p* < 0.001	*H*(4) = 53.343 * *p* < 0.001	*H*(4) = 38.717 * *p* < 0.001	*H*(4) = 19.330 * *p* < 0.001
Single	14.6 (62)	23.53 ± 5.45 (24.0 ± 9.0)	3.69 ± 1.06 (4.0 ± 1.0)	3.77 ± 0.80 (4.0 ± 1.0)	3.02 ± 0.66 (3.0 ± 0.74)
Married	41.8 (178)	25.24 ± 5.54 (27.0 ± 8.0)	3.56 ± 0.96 (4.0 ± 1.0)	3.83 ± 0.71 (4.0 ± 1.0)	3.38 ± 0.94 (3.74 ± 1.0)
Divorced	7 (30)	23.13 ± 4.38 (23.0 ± 7.5)	2.70 ± 1.18 (3.0 ± 1.0)	3.80 ± 0.41 (4.0 ± 1.0)	3.39 ± 0.63 (3.26 ± 0.94)
Cohabitation	24.6 (105)	24.79 ± 5.74 (27.0 ± 9.0)	3.50 ± 0.92 (4.0 ± 0.0)	3.95 ± 0.59 (4.0 ± 1.0)	3.30 ± 0.73 (3.0 ± 0.65)
Widowed	12 (51)	21.04 ± 6.23 (22.0 ± 5.0)	2.73 ± 0.75 (3.0 ± 1.0)	3.20 ± 0.80 (3.0 ± 1.0)	3.49 ± 0.81 (3.42 ± 1.16)
Education % (*n*)	—	*H*(3) = 22.590 * *p* < 0.001	*H*(3) = 23.174 * *p* < 0.001	*H*(3) = 36.689 * *p* < 0.001	*H*(3) = 3.719 * *p* = 0.293
Elementary education	9.2 (39)	21.36 ± 7.08 (21.0 ± 11.0)	2.87 ± 1.22 (3.0 ± 1.0)	3.54 ± 0.97 (4.0 ± 1.0)	3.47 ± 0.74 (3.55 ± 1.03)
Secondary Education	51.9 (221)	23.68 ± 5.79 (24.0 ± 8.0)	3.34 ± 0.97 (3.0 ± 1.0)	3.62 ± 0.72 (4.0 ± 1.0)	3.32 ± 0.85 (3.20 ± 1.0)
HE (Bachelor)	30 (128)	27.27 ± 5.56 (27.0 ± 9.57)	3.70 ± 0.77 (4.0 ± 1.0)	4.04 ± 0.57 (4.0 ± 0.0)	3.29 ± 0.85 (3.0 ± 0.99)
HE (Master or Doctoral)	8.9 (38)	26.80 ± 4.63 (28.0 ± 6.0)	3.26 ± 1.41 (4.0 ± 1.0)	4.03 ± 0.59 (4.0 ± 0.0)	3.30 ± 0.66 (3.16 ± 0.65)
No. of NCD % (*n*)	—	*H*(1) = 0.105 *p* = 0.745	*H*(1) = 0.276 *p* = 0.599	*H*(1) = 10.431 * *p* < 0.001	*H*(1) = 0.691 *p* = 0.496
One	57.7 (246)	24.32 ± 5.97 (25.0 ± 8.0)	3.39 ± 1.02 (4.0 ± 1.0)	3.69 ± 0.76 (4.0 ± 1.0)	3.35 ± 0.91 (3.22 ± 1.0)
Two to three	40.1 (171)	24.26 ± 5.44 (25.0 ± 9.0)	3.44 ± 0.98 (4.0 ± 1.0)	3.92 ± 0.57 (4.0 ± 0.0)	3.31 ± 0.67 (3.10 ± 0.95)
Four or more	2.1 (09)	21.11 ± 10.83 (25 ± 20)	2.89 ± 1.36 (2.0 ± 2.0)	3.44 ± 1.51 (3.0 ± 3.0)	3.00 ± 0.85 (3.0 ± 0.98)
Managing NCD % (*n*)	—	*H*(3) = 18.125 * *p* < 0.001	*H*(3) = 45.364 * *p* < 0.001	*H*(3) = 33.386 * *p* < 0.001	*H*(3) = 11.622 * *p* = 0.009
Very poor	7.5 (32)	25.94 ± 4.22 (27.0 ± 6.5)	2.81 ± 1.26 (3.0 ± 1.0)	3.88 ± 0.55 (4.0 ± 2.0)	3.49 ± 0.72 (3.42 ± 0.74)
Poor	9.2 (39)	20.56 ± 5.62 (22.0 ± 8.0)	2.64 ± 0.71 (3.0 ± 1.0)	3.15 ± 0.75 (3.0 ± 1.0)	3.21 ± 0.91 (3.0 ± 1.0)
Good	75.4 (321)	24.39 ± 6.01 26.0 ± 9.0)	3.52 ± 0.96 (4.0 ± 1.0)	3.84 ± 0.69 (4.0 ± 0.0)	3.28 ± 0.82 (3.16 ± 0.96)
Very good	8.0 (34)	25.35 ± 5.01 (26.0 ± 7.5)	3.68 ± 1.01 (4.0 ± 1.0)	3.74 ± 0.83 (4.0 ± 1.0)	3.70 ± 0.79 (3.69 ± 1.30)
Contacts with loved ones % (*n*)	—	*H*(3) = 4.779 *p* = 0.189	*H*(3) = 9.300 * *p* = 0.026	*H*(3) = 5.846 *p* = 0.119	*H*(3) = 3.455 *p* = 0.327
Less than once/month	5.6 (24)	25.33 ± 3.75 24.0 ± 5.75)	2.88 ± 0.99 (3.0 ± 2.0)	3.83 ± 0.70 (4.0 ± 0.0)	3.56 ± 0.83 (3.73 ± 1.03)
Once/week to once/month	23.2 (99)	23.25 ± 5.74 (23.0 ± 9.0)	3.43 ± 0.94 (3.0 ± 1.0)	3.62 ± 0.78 (4.0 ± 1.0)	3.27 ± 0.71 (3.11 ± 0.96)
Few timed/week	31.9 (136)	24.04 ± 6.91 (26.0 ± 9.0)	3.50 ± 1.01 (4.0 ± 1.0)	3.83 ± 0.69 (4.0 ± 0.0)	3.29 ± 0.99 (3.05 ± 1.01)
Daily	39.2 (167)	24.81 ± 5.27 (25.0 ± 8.0)	3.81 ± 0.71 (4.0 ± 1.0)	3.81 ± 0.71 (4.0 ± 1.0)	3.35 ± 0.74 (3.20 ± 0.85)
Living environment	—	*U* = 10,202.5 * *p* < 0.001	*U* = 9566.0 * *p* < 0.001	*U* = 10,493.0 * *p* < 0.001	*U* = 13,612.0 * *p* < 0.001
Home environment	—	3.86 ± 0.67 (4 ± 1)	3.51 ± 0.98 (4 ± 1)	24.66 ± 6.05 (26 ± 8.25)	3.33 ± 0.84 (3.14 ± 1)
Nursing home	—	3.43 ± 0.81 (4 ± 1)	2.94 ± 1.03 (3 ± 2)	22.46 ± 4.88 (23 ± 6.75)	3.29 ± 0.76 (3.18 ± 0.95)
Physical activity % (*n*)	—	*H*(4) = 6.827 *p* = 0.078	*H*(4) = 23.667 * *p* < 0.001	*H*(4) = 27.186 * *p* < 0.001	*H*(4) = 8.153 * *p* = 0.043
Rarely	5.9 (25)	21.88 ± 9.03 (24.0 ± 15.5)	3.12 ± 1.33 (3.0 ± 2.0)	3.56 ± 0.92 (4.0 ± 1.0)	2.76 ± 0.97 (3.0 ± 0.75)
Very rare (1–4/month)	12 (51)	22.75 ± 5.09 (23.0 ± 7.0)	3.18 ± 0.84 (3.0 ± 1.0)	3.53 ± 0.67 (4.0 ± 1.0)	3.17 ± 0.88 (3.18 ± 1.13)
Rare (5–10/month)	41.3 (176)	24.24 ± 6.50 (24.5 ± 8.0)	3.00 ± 1.10 (3.0 ± 2.0)	3.59 ± 0.68 (4.0 ± 1.0)	3.36 ± 0.67 (3.11 ± 0.88)
Quite often (2.4/week)	24.4 (104)	24.44 ± 6.50 (26.0 ± 8.75)	3.60 ± 0.95 (4.0 ± 1.0)	3.92 ± 0.76 (4.0 ± 0.0)	3.36 ± 0.87 (3.18 ± 1.0)
Very often (5 or more/week)	16.4 (70)	25.60 ± 6.99 (27.0 ± 8.25)	3.77 ± 0.75 (4.0 ± 1.0)	3.94 ± 0.51 (4.0 ± 0.0)	3.49 ± 0.71 (3.50 ± 1.02)

Note: n—Sample size; %—Percent of participants; *—Statistical significance (*p* < 0.05); SD—Standard deviation; M—Mean; Me—Median; IQR—Interquartile range; HE—Higher Education; NCD—non-communicable disease; U—Mann–Whitney U test; H—Kruskal–Wallis test.

## Data Availability

Additional data from this study are not publicly available to maintain participants’ anonymity.

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
