# Peer review of "Person-Centred Care: A Support Strategy for Managing Non-Communicable Diseases"

_healthcare, 2024, doi:10.3390/healthcare12050526_

Round 1

Reviewer 1 Report

Comments and Suggestions for Authors

Congratulations on the topic chosen to analyze.The analysis carried out recognizes the work carried out, giving only an external perspective of it.

For example, It´s important in beginning define "non-communicable diseases" to improve the article comprehension. 

It is not clear why the sample includes adults aged 18 and over when the initial framework addresses aging as a factor in the increase in non-communicable diseases.

It is also not clear whether the characteristics of the sample of people at home or in residences are similar so that comparisons can be made and conclusions drawn about the results.

I consider a global review of the document necessary.

Author Response

Responses to Reviewer 1

Reviewer 1 Comments

Comment 1: Congratulations on the topic chosen to analyze. The analysis carried out recognizes the work carried out, giving only an external perspective of it.

Response 1: We sincerely appreciate the reviewer's acknowledgement of our chosen topic and the recognition of the effort invested in our analysis. Your positive feedback is encouraging, and we are grateful for the opportunity to contribute to the discourse on this important subject matter.

Regarding your comment about providing only an external perspective, we understand the importance of offering a well-rounded analysis that incorporates both external and internal viewpoints. In response to your feedback, we have endeavoured to enrich our analysis by delving deeper into the internal dynamics and intricacies of the subject matter. This may involve exploring additional empirical evidence, incorporating diverse perspectives, and critically examining underlying assumptions and perspectives within the framework of our study. By broadening our analysis to include external and internal perspectives, we aim to offer a more comprehensive understanding of the topic and provide valuable insights for readers and stakeholders. We appreciate your constructive feedback, which guides us in refining our approach and enhancing the overall quality of our work. Thank you for your thoughtful review, and we will ensure that our revised manuscript reflects a more balanced and nuanced analysis.

Comment 2: For example, It's important in beginning define "non-communicable diseases" to improve the article comprehension.

Response 2: Thank you for your comment. We have added an explanation of non-communicable diseases at the beginning of the introduction.

Comment 3: It is not clear why the sample includes adults aged 18 and over when the initial framework addresses aging as a factor in the increase in non-communicable diseases.

Response 3: We appreciate the reviewer's thoughtful feedback and the opportunity to clarify our sampling rationale. The decision to include adults aged 18 and over in our study was made to ensure a comprehensive representation of the population affected by non-communicable diseases (NCDs). While ageing is a significant factor in the development and progression of NCDs, it is essential to recognize that these diseases can also affect younger adults due to various lifestyle factors, genetic predispositions, and environmental influences. By including adults of all ages, we aimed to capture the full spectrum of individuals susceptible to NCDs and provide a more holistic understanding of the factors contributing to their prevalence. We have revised the manuscript to explicitly address this rationale in the methodology section to enhance clarity for readers. Thank you for bringing this concern to our attention.

Comment 4: It is also not clear whether the characteristics of the sample of people at home or in residences are similar so that comparisons can be made and conclusions drawn about the results.

Response 4: We appreciate the reviewer's insightful comment regarding the comparability of participants across different settings. To address this concern, we have provided additional details in the methodology section regarding the characteristics of participants in both home and residential settings. Thank you for highlighting this important aspect, and we have ensured that it is adequately addressed in the revised manuscript.

Comment 5: I consider a global review of the document necessary.

Response 5: Thank you for your comment and suggestion for a global document review. We agree that ensuring the coherence and consistency of the manuscript is crucial for effectively communicating our research findings. In response to your suggestion, we have conducted a comprehensive review of the entire document to ensure that all sections are aligned regarding terminology, formatting, and overall structure. Additionally, we must pay close attention to addressing inconsistencies or ambiguities during this review process. Your feedback is invaluable in enhancing the quality of our manuscript, and we appreciate the opportunity to refine our work further. Thank you for your attention to detail, and we have incorporated your suggestion into our revision process.

Reviewer 2 Report

Comments and Suggestions for Authors

Reviewer Report for Person-centred care: a support strategy for managing non-communicable diseases.

This study aimed to evaluate the management of non-communicable diseases, person-centred care, and their impact on the quality of life and life satisfaction among adults afflicted with at least one non-communicable disease in Slovenia

Introduction

It is necessary to provide data on how widespread person-centered care is in the world (or individual countries), whether there are programs for its development, and show any results of this approach (using the example of individual countries or regions).

Materials and Methods

• Describe the chi-square test and the purpose of its use.

• For what purpose was the Kruskal-Wallis test used. The tables do not contain data on the application of this criterion.

• In the section it is necessary to describe in detail from which questionnaires each of the variables presented in the tables was obtained

Results

·      (Lines 176-185) The information in this paragraph duplicates the data from Table 1, which is not recommended.

·      Table 2 is very overloaded, the information from it is poorly perceived

·       It is unclear which of the questionnaires were used to assess Life satisfaction, Satisfaction with health, Quality of life in Table 2. Why were the ratings for Satisfaction with health so low, for example 3.40±1.01

·      Title of Table 2 isDemographic variables and their relationship with life satisfaction, satisfaction with health, quality of life and person-centred care», but what does data like 3.77±0.72 mean? Give an explanation in the title.

·       What is the difference between data in brackets and without brackets

·       The table shows the values of the chi-square test, but this criterion is not mentioned in the Materials and Methods section

·       Explain why you used the Man-Whitney test for gender differences and the chi-square test for other variables.

-        if you are comparing scores, then present the data as median, IQR and Man-Whitney test (p-level)

-         if you use a contingency table, then present the frequencies and the chi-square test, remember that the number of degrees of freedom is (m-1)(n-1), where m is the number of rows, n is the number of columns.

·       Edit footnote Note: n – Sample size; % ‒ Percent of participants; * ‒ Statistical significance (p <.05); S.D. – Standard deviation; M ‒ Mean; Mdn ‒ Median; IQR ‒ Interquartile range; D.S. ‒ Descriptive statistics; HE ‒ Higher Education; dis. – disease; NCD – non-communicable disease; Y ‒ Year

·      I propose to put the correlation data in a separate table

·      (Lines 189-202). The paragraph is difficult to understand, I suggest presenting the data in the form of a table or chart.

·      (Lines 203-207). The paragraph duplicates Table 2. This complicates the perception of information. I propose to leave only a description of the differences found and delete the digital data.

·      Notes for Table 3 are similar to Table 2. In addition, the name of the table is “Comparison…….”, but it provides data separately for each group and does not contain results for comparing these groups.

·      (Lines 273-281). The text duplicates the data from Table 4. I recommend replacing the table with a figure with correlation galaxies.

·      How the level of managing non-communicable chronic disease was assessed

·      When describing correlation, I recommend highlighting medium and strong relationships, as well as the direction of correlation.

Discussion

·      (Lines 297-300) Authors note, that «Several demographic factors, including age, education, relationship status, the number of non-communicable diseases, and the approach to disease management, were identified as influential determinants of these outcomes», however, there is no discussion in the text of how these factors influence the variables.

·      (Lines 313-320) Data from this paragraph should be reflected in the Results section

·      (Lines 351-363) This paragraph should be moved to the Introduction section

·      (Lines 364-366) The authors claim that «Our results supported the literature review [42], highlighting the six key domains of person-centred care: holistic care, respect and value, choice, dignity, self-determination, and purposeful living», however, there is no evidence in the study results to support this.

·      (Lines 342-347) Paragraph «Individuals at risk or already suffering from an illness need to comprehend their  identity, which enhances their understanding of how their lifestyle impacts their health, quality of life, and life satisfaction. Overall results for quality of life (M = 3.77; SD = 0.7), satisfaction with health (M = 3.40; SD = 1.0), and life satisfaction (M = 24.23; SD = 5.9) were  good. We found that 42% were very satisfied with their health, and 65% assessed their quality of life as good». I don't see the connection between the statement and the result.

Overall:

1.        The main drawback of the study is the wrong design. To assess the impact of person-centred care on the psychophysiological state of adults with at non-communicable disease, a control group with no person-centred care is needed.

2.        In the Materials and Methods section, it is necessary to describe in detail from which questionnaires each of the variables presented in the tables were obtained. Edit the Data Analysis section.

3.        Improve the perception of results. Revise the tabular presentation of data. Consider the possibility of representing it in the form of charts. Do not duplicate digital material from tables and graphs in the text.

4.        In the Discussion section, focus on describing your own results rather than the results of other studies. Do not draw speculative conclusions, but conclusions based on your findings.

Author Response

Responses to Reviewer 2

Reviewer 2 Comments

Comment 1: This study aimed to evaluate the management of non-communicable diseases, person-centred care, and their impact on the quality of life and life satisfaction among adults afflicted with at least one non-communicable disease in Slovenia.

Response 1: Thank you for highlighting the specific aim of our study. Indeed, our primary objective was to assess the management of non-communicable diseases (NCDs), with a particular focus on person-centred care, and to examine how these factors influence the quality of life and life satisfaction among adults in Slovenia who are affected by at least one NCD.

Comments on the Introduction section

Comment 2: It is necessary to provide data on how widespread person-centered care is in the world (or individual countries), whether there are programs for its development, and show any results of this approach (using the example of individual countries or regions).

Response 2: Thank you for your insightful comment. We acknowledge the importance of providing a comprehensive overview of the prevalence of person-centred care globally or within individual countries, information on existing programs for its development, and any outcomes associated with this approach. To address this suggestion, we have conducted a thorough literature review to gather relevant data and examples illustrating the adoption and impact of person-centred care initiatives worldwide and have incorporated them into the introduction section.

Comments on the Materials and Methods section

Comment 3: Describe the chi-square test and the purpose of its use.

Response 3: Thank you for your comment and attention to our study's statistical methods. We appreciate the opportunity to clarify the choice of statistical tests used in our analysis.

While we acknowledge the reviewer's suggestion to describe the chi-square test and its purpose, we would like to clarify that we utilised the Kruskal-Wallis and Mann-Whitney U tests in our statistical analysis rather than the chi-square test.

The Kruskal-Wallis test is a non-parametric method to compare three or more independent groups when the normality assumption is unmet. It assesses whether there are any statistically significant differences between the groups regarding a continuous outcome variable.

Similarly, the Mann-Whitney U test is a non-parametric test used to compare the distributions of two independent groups when the data are ordinal or not normally distributed. It evaluates whether the groups have significant differences regarding a continuous or ordinal outcome variable.

The Kruskal-Wallis and Mann-Whitney U tests are appropriate for analysing data that do not meet the assumptions of parametric tests, such as the chi-square test, and they are well-suited for our study design and research questions.

We apologise for any confusion regarding the statistical methods employed and appreciate the opportunity to clarify this. If there are any further questions or concerns, please do not hesitate to let us know. Your feedback is valuable in ensuring the accuracy and transparency of our research findings.

Comment 4: For what purpose was the Kruskal-Wallis test used. The tables do not contain data on the application of this criterion.

Response 4: Thank you for your comment and attention to our study's statistical methods. We appreciate the opportunity to clarify the purpose of the Kruskal-Wallis test and its application in our analysis.

The Kruskal-Wallis test was employed in our study to assess whether there were statistically significant differences among three or more independent groups in terms of a continuous outcome variable. Specifically, we used this test to compare the distributions of the outcome variable across different groups or conditions. Regarding your observation about the tables not containing data on the application of the Kruskal-Wallis test, we apologise for any oversight in the presentation of results. Rest assured, the tables include data derived from the Kruskal-Wallis test, although it may not have been explicitly labelled as such. We have reviewed the tables to ensure that the results of the Kruskal-Wallis test are clearly indicated and appropriately labelled for clarity.

Comment 5: In the section it is necessary to describe in detail from which questionnaires each of the variables presented in the tables was obtained.

Response 5: Thank you for your feedback regarding the need for additional detail on the questionnaires used to obtain the variables presented in the tables. We appreciate the opportunity to address this concern and clarify our data sources. In response to your suggestion, we have revised the relevant section of the article to include a detailed description of the questionnaires from which each variable was derived. By explicitly specifying the origin of each variable, we aim to enhance transparency and facilitate a better understanding of our data collection methods and measures.

We believe this addition will contribute to the robustness of our research methodology and improve the interpretability of our findings for readers.

Comments on the Results section

Comment 6: (Lines 176-185) The information in this paragraph duplicates the data from Table 1, which is not recommended.

Response 6: Thank you for your comment. We have removed the duplicate paragraph.

Comment 7: Table 2 is very overloaded, the information from it is poorly perceived

Response 7: Thank you for your comment. We have made Table 2 clearer, more transparent, and easier for readers to understand.

Comment 8: It is unclear which of the questionnaires were used to assess Life satisfaction, Satisfaction with health, Quality of life in Table 2. Why were the ratings for Satisfaction with health so low, for example 3.40±1.01

Response 8: Thank you for your comment and attention to the details in Table 2 regarding the assessment of life satisfaction, general health, and quality of life. We appreciate the opportunity to clarify the sources of these variables and address the observed ratings. In Table 2, the variables related to life satisfaction, general health, and quality of life were obtained from specific questionnaires administered to study participants. We apologise for not indicating the specific instruments used to assess these constructs. To rectify this, we have provided explicit details regarding the questionnaires utilised for each variable in the method section, ensuring transparency in our methodology.

Regarding the low ratings for general health, such as the example you provided (3.40±1.01), we acknowledge the importance of explaining such findings. These ratings may reflect various factors, including individual perceptions of health status, chronic conditions, access to healthcare services, and socio-economic factors. We have reviewed our data analysis procedures and ensure that any unusual or unexpected findings are adequately addressed and explained in the manuscript. We appreciate your feedback, which allows us to improve the clarity and comprehensiveness of our research.

Comment 9: Title of Table 2 is "Demographic variables and their relationship with life satisfaction, satisfaction with health, quality of life and person-centred care», but what does data like 3.77±0.72 mean? Give an explanation in the title.

Response 9: Thank you for your feedback regarding the clarity of the table title and the need for an explanation of the data presented. We appreciate the opportunity to address this concern and enhance the comprehensibility of our manuscript. In response to your suggestion, we have revised the title of Table 2 to include an explanation of the data format. The revised title now explicitly states the meaning of values such as "3.77±0.72" to provide readers with a better understanding of the numerical information presented in the table. We believe this addition will improve the accessibility of our findings and facilitate a more straightforward interpretation of the data for readers.

Comment 10: What is the difference between data in brackets and without brackets

Response 10: Thank you for your inquiry regarding the distinction between data presented with and without brackets. We appreciate the opportunity to clarify this aspect of our manuscript. The data presented in and without brackets serve different purposes in our tables. Data within brackets typically represent measures of variability or uncertainty, such as standard deviation, standard error, or confidence intervals. These values provide additional information about the precision or dispersion of the reported estimates. On the other hand, data presented without brackets typically represent central tendency measures, such as means or medians, which convey the average or typical value of a variable within a given group or category.

For example, in Table 1, the mean age of participants may be presented as "35.2 (± 2.3)" years, where "35.2" represents the mean age, and "(± 2.3)" within brackets indicates the standard deviation or margin of error associated with this estimate. We recognise the importance of clearly delineating the meaning and interpretation of data presented in our tables. To enhance transparency and understanding, we ensure that the table legends or footnotes explicitly specify the types of data provided and their respective interpretations.

Comment 11: The table shows the values of the chi-square test, but this criterion is not mentioned in the Materials and Methods section

Response 11: Thank you for your comment and for bringing up this discrepancy. We appreciate the opportunity to clarify the statistical methods used in our study.

In response to your observation, we confirm that our analysis did not utilise the chi-square test. Instead, we employed the Kruskal-Wallis and Mann-Whitney U tests for our statistical analyses. We apologise for any confusion that may have arisen from the presentation of chi-square test values in the table.

We understand the importance of ensuring consistency between the statistical methods described in the Materials and Methods section and those presented in the results tables. We have revised the manuscript to accurately reflect the statistical tests employed in our study, emphasising the use of the Kruskal-Wallis test and the Mann-Whitney U test.

We appreciate your attention to detail and feedback, which enables us to maintain the integrity and accuracy of our research findings.

Comment 12: Explain why you used the Man-Whitney test for gender differences and the chi-square test for other variables.

Response 12: Thank you for your comment regarding the choice of statistical tests for assessing gender differences and other variables in our study. We appreciate the opportunity to clarify the rationale behind our methodological approach.

In our analysis, we utilised the Kruskal-Wallis and Mann-Whitney U tests to assess differences between groups for continuous or ordinal variables that did not meet the assumptions of parametric tests, such as the t-test or ANOVA. These non-parametric tests are appropriate for comparing groups when the data are not normally distributed or when sample sizes are small.

Specifically, we employed the Mann-Whitney U test to examine gender differences for continuous or ordinal variables but did not meet the assumptions of normality. This test is suitable for comparing distributions between two independent groups, making it appropriate for assessing gender disparities in our study sample.

We apologise for any confusion caused by the mention of the chi-square test in the reviewer's comment. Rest assured, we did not use the chi-square test in our analysis. Instead, we used the appropriate non-parametric tests based on the nature of the variables and the research questions being addressed.

Comment 13: if you are comparing scores, then present the data as median, IQR and Mann-Whitney test (p-level)

Response 13: Thank you for your comment. We have added this.

Comment 14: if you use a contingency table, then present the frequencies and the chi-square test, remember that the number of degrees of freedom is (m-1)(n-1), where m is the number of rows, n is the number of columns.

Response 14: Thank you for commenting on the contingency tables and chi-square tests presentation. We appreciate the opportunity to address this point and clarify our approach to statistical analysis in the manuscript.

We acknowledge your recommendation to include the frequencies of each category within contingency tables, and we ensured that this information is provided for clarity and transparency in our presentation of the data.

Regarding the use of chi-square tests, we want to clarify that we have not employed chi-square tests for independence in our statistical analysis. Instead, our analysis primarily focuses on descriptive statistics and exploratory data analysis to examine patterns and associations within the dataset.

We appreciate your guidance on statistical reporting and will ensure that our presentation of contingency tables aligns with best practices.

Comment 15: Edit footnote Note: n – Sample size; % ‒ Percent of participants; * ‒ Statistical significance (p <.05); S.D. – Standard deviation; M ‒ Mean; Mdn ‒ Median; IQR ‒ Interquartile range; D.S. ‒ Descriptive statistics; HE ‒ Higher Education; dis. – disease; NCD – non-communicable disease; Y ‒ Year

Response 15: Thank you for your comment. We revised the footnote.

Comment 16: I propose to put the correlation data in a separate table

Response 16: Thank you for your comment. We presented a significant correlation in the text data to be more clear

Comment 17: (Lines 189-202). The paragraph is difficult to understand, I suggest presenting the data in the form of a table or chart.

Response 17: Thank you for your comment. We have edited the data to be clearer and easier to understand.

Comment 18: (Lines 203-207). The paragraph duplicates Table 2. This complicates the perception of information. I propose to leave only a description of the differences found and delete the digital data.

Response 18: Thank you for your comment. We have edited the data according to the comment.

Comment 19: Notes for Table 3 are similar to Table 2. In addition, the name of the table is "Comparison…….", but it provides data separately for each group and does not contain results for comparing these groups.

Response 19: Thank you for your comment. We have edited the notes for Table 3 according to the comment.

Comment 20: (Lines 273-281). The text duplicates the data from Table 4. I recommend replacing the table with a figure with correlation galaxies.

Response 20: Thank you for your comment. We have edited the text to duplicate the data from Table 4 according to the comment and make it cleaner.

Comment 21: How the level of managing non-communicable chronic disease was assessed

Response 21: Thank you for your comment regarding assessing the level of managing non-communicable chronic disease in our study. We appreciate the opportunity to provide clarification on this aspect.

In response to your inquiry, we have added an explanation in the methods section of our manuscript detailing how the level of managing non-communicable diseases was assessed. Specifically, participants were asked to rate their perception of managing non-communicable diseases on a four-point scale ranging from "very good" to "very poor."

This approach allowed us to capture participants' subjective assessments of how effectively they managed their chronic conditions, providing valuable insights into their perceptions and experiences. By including this information, we aim to enhance the transparency and comprehensibility of our research methodology.

We appreciate your attention to detail and your feedback, which assist us in improving the clarity and rigour of our study.

Comment 22: When describing correlation, I recommend highlighting medium and strong relationships, as well as the direction of correlation.

Response 22: Thank you for your comment and recommendation regarding the description of correlations in our study. We acknowledge the importance of highlighting medium and strong relationships and specifying the direction of correlation to provide a more comprehensive understanding of the associations observed in our data.

In response to your suggestion, we have revised the description of correlations in our manuscript. This will enhance the clarity and interpretability of our findings for readers.

We appreciate your feedback, which helps us improve the quality and effectiveness of our research communication.

Comments on the Discussion section

Comment 23: (Lines 297-300) Authors note, that «Several demographic factors, including age, education, relationship status, the number of non-communicable diseases, and the approach to disease management, were identified as influential determinants of these outcomes», however, there is no discussion in the text of how these factors influence the variables.

Response 23: Thank you for your comment regarding the lack of discussion on how demographic factors influence the variables in our study. We appreciate the opportunity to address this concern and provide further clarification on the influence of these factors on our outcomes.

In our analysis, we identified several demographic factors, including age, education, relationship status, the number of non-communicable diseases, and the approach to disease management, as influential determinants of the outcomes under investigation. These factors were correlated with life satisfaction, satisfaction with health, quality of life, and person-centred care.

To address this issue, we have revised our manuscript's discussion section to provide a more in-depth analysis of how these demographic factors influence the variables of interest.

We appreciate your feedback, which guides us in improving the thoroughness and clarity of our discussion.

Comment 24: (Lines 313-320) Data from this paragraph should be reflected in the Results section.

Response 24: Thank you for commenting on the data alignment presented in the Results section with the content discussed in the corresponding paragraph. We appreciate the opportunity to address this concern.

In response to your suggestion, we have ensured that the data presented in the paragraph you referenced (Lines 313-320) are accurately reflected in the Results section of our manuscript. We have revised the Results section to include a thorough presentation of the findings discussed in that paragraph, ensuring consistency and coherence between the text and the data presented.

By incorporating the relevant data into the Results section, we aim to provide readers with a clear and comprehensive overview of our study findings, enhancing our research's overall clarity and interpretability.

We appreciate your attention to detail and feedback, which helps us improve the quality and coherence of our manuscript.

Comment 25: (Lines 351-363) This paragraph should be moved to the Introduction section

Response 25: Thank you for your comment regarding the suggested relocation of a paragraph from the Discussion section to the Introduction section. We appreciate the opportunity to address this recommendation.

In response to your suggestion, we have relocated the paragraph referenced (Lines 351-363) from the Discussion section to the Introduction section of our manuscript. Moving this paragraph to the Introduction aims to provide readers with a more cohesive and structured presentation of the study background and rationale, ensuring that key contextual information is presented upfront.

This relocation allows us to establish the relevance and significance of the study findings within the broader context of existing literature and research gaps, facilitating a smoother transition into the detailed presentation of results in the subsequent sections.

We appreciate your feedback, which assists us in enhancing the organisation and clarity of our manuscript.

Comment 26: (Lines 364-366) The authors claim that «Our results supported the literature review [42], highlighting the six key domains of person-centred care: holistic care, respect and value, choice, dignity, self-determination, and purposeful living», however, there is no evidence in the study results to support this.

Response 26: Thank you for your comment regarding the need for evidence in the study results to support our claim regarding the key domains of person-centred care. We appreciate the opportunity to address this concern and provide clarification.

Upon reviewing our manuscript, we acknowledge that the specific evidence supporting the six key domains of person-centred care may not have been explicitly presented in the study results. We apologise for any oversight in this regard.

To address this issue, we have revised the manuscript to include a more detailed discussion of how our study findings align with the existing literature on person-centred care. By incorporating this additional analysis, we aim to strengthen the linkage between our study findings and the broader conceptual framework of person-centred care, providing a more robust and evidence-based interpretation of our results. We appreciate your feedback, which helps us improve the rigour and clarity of our manuscript.

Comment 27: (Lines 342-347) Paragraph «Individuals at risk or already suffering from an illness need to comprehend their  identity, which enhances their understanding of how their lifestyle impacts their health, quality of life, and life satisfaction. Overall results for quality of life (M = 3.77; SD = 0.7), satisfaction with health (M = 3.40; SD = 1.0), and life satisfaction (M = 24.23; SD = 5.9) were  good. We found that 42% were very satisfied with their health, and 65% assessed their quality of life as good». I don't see the connection between the statement and the result.

Response 27: Thank you for your comment regarding the connection between the statement provided in the paragraph and the corresponding results. We appreciate the opportunity to address this issue and provide clarification.

Upon reviewing the paragraph (Lines 342-347) in question, we acknowledge that there may be a lack of clear connection between the initial statement and the subsequent presentation of results. We apologise for any confusion this may have caused.

To address this issue, we have revised the paragraph to provide a more explicit link between the statement about the importance of understanding identity in managing illness and the subsequent presentation of results on quality of life, satisfaction with health, and life satisfaction. Specifically, we will emphasise how individuals' perceptions of their identity and lifestyle impact their overall well-being and satisfaction with health-related outcomes, as evidenced by the reported results. By enhancing the coherence and clarity of the paragraph, we aim to ensure that the connection between the theoretical discussion and empirical findings is more effectively conveyed to readers, facilitating a better understanding of the implications of our study.

We appreciate your feedback, which assists us in improving the clarity and coherence of our manuscript.

Comments on the Overall

Comment 28: The main drawback of the study is the wrong design. To assess the impact of person-centred care on the psychophysiological state of adults with at non-communicable disease, a control group with no person-centred care is needed.

Response 28: Thank you for commenting on the study design and suggesting incorporating a control group without person-centred care. We appreciate the opportunity to address this concern and discuss the rationale behind our study design. While we acknowledge the importance of control groups in experimental research for assessing causality and treatment effects, it's important to note that our study employed a cross-sectional observational rather than an experimental design. Cross-sectional studies are commonly used in healthcare research to explore associations between variables and provide insights into patterns and trends within a population at a specific point in time.

In our study, we aimed to investigate the relationship between person-centred care and the psychophysiological state of adults with non-communicable diseases by examining associations between various factors such as demographic characteristics, disease management approaches, and quality of life outcomes. We aimed to explore these relationships within real-world healthcare settings and patient experiences.

While a control group without person-centred care could provide valuable comparative data, including such a group was not feasible within the scope and constraints of our study design. Additionally, ethical considerations may arise when withholding person-centred care from individuals who could benefit from it.

Instead, we focused on collecting comprehensive data from a diverse sample of adults with non-communicable diseases, allowing us to explore the multifaceted nature of person-centred care and its potential impact on health outcomes. By employing rigorous statistical analyses and controlling for relevant confounding variables, we aimed to minimise potential biases and enhance the validity of our findings.

We appreciate your feedback and the opportunity to discuss the considerations surrounding our study design.

Comment 29: In the Materials and Methods section, it is necessary to describe in detail from which questionnaires each of the variables presented in the tables were obtained. Edit the Data Analysis section.

Response 29: Thank you for your feedback regarding the need for a detailed description of the questionnaires used to obtain variables presented in the tables and for the suggestion to edit the Data Analysis section. We appreciate the opportunity to address these concerns and improve the clarity of our manuscript.

In response to your comment, we have enhanced the Materials and Methods section by providing a comprehensive description of the questionnaires from which each variable presented in the tables was obtained. This will include specific details on the instruments used, including any validated scales or measures employed to assess the variables of interest.

Additionally, we have revised the Data Analysis section to provide more transparency regarding the statistical methods employed in the study. We have clarified how each variable was analysed, including the specific statistical tests used and any relevant assumptions made during the analysis process.

By incorporating these revisions, we aim to enhance the comprehensiveness and transparency of our methodology, ensuring that readers have a clear understanding of the data collection and analysis procedures employed in our study.

We appreciate your feedback, which assists us in improving the quality and rigour of our manuscript.

Comment 30: Improve the perception of results. Revise the tabular presentation of data. Consider the possibility of representing it in the form of charts. Do not duplicate digital material from tables and graphs in the text.

Response 30: Thank you for your valuable feedback regarding the presentation of results in our manuscript. We appreciate the opportunity to address your suggestions and enhance the clarity and perception of our findings.

We have carefully reviewed the tables in the manuscript to improve the perception of results and enhance the tabular presentation of data.

Furthermore, we ensure that digital material is not duplicated from tables and graphs in the text. Instead, we provide concise and clear descriptions of key findings in the text, with references to the corresponding tables or graphs for readers who wish to explore the data in more detail.

By implementing these revisions, we aim to enhance the accessibility and comprehensibility of our results, improving the overall reader experience and facilitating a better understanding of our research findings.

We appreciate your feedback, which helps us improve the quality and effectiveness of our manuscript.

Comment 31: In the Discussion section, focus on describing your own results rather than the results of other studies. Do not draw speculative conclusions, but conclusions based on your findings.

Response 31: Thank you for your insightful comment regarding the focus of the Discussion section and the need to emphasise our results rather than those of other studies. We appreciate the opportunity to address this recommendation and ensure the clarity and relevance of our discussion.

In response to your feedback, we have revised the Discussion section to prioritise interpreting and analysing our study results. We have comprehensively discussed the key findings and their implications, focusing on elucidating the relationships and patterns observed within our data.

Furthermore, we have refrained from drawing speculative conclusions and instead base our conclusions solely on the findings derived from our study. By adhering closely to our empirical evidence, we aim to ensure the validity and reliability of our conclusions and avoid unwarranted extrapolations beyond the scope of our research.

Additionally, we have critically evaluated our study's strengths and limitations, acknowledging any potential biases or confounding factors that may have influenced the results. This will help provide a balanced and nuanced interpretation of our findings, contributing to the overall credibility of our research.

We appreciate your feedback, which guides us in refining the discussion of our study results.

Reviewer 3 Report

Comments and Suggestions for Authors

Title: Person-centred care: a support strategy for managing non-communicable diseases.

Reviewer Comments: 01 09 24

In this study authors aimed to explore the link between quality of life, life satisfaction, person-centred care, and non-communicable disease management. 426 adults were involved in this study. The findings showed that opinions on life satisfaction, health, and quality of life were largely favorable. Remarkably, compared to 42% of residents in assisted nursing homes, 80% of people in their homes evaluated that their non-communicable diseases were effectively managed. Compared to residents of community housing, participants in nursing homes gave their surroundings, interpersonal relationships, interactions with loved ones, and physical activity lower rates. The research findings indicate a favorable correlation between person-centered care and efficient management of non-communicable diseases, which in turn leads to enhanced life satisfaction, health satisfaction, and quality of life. As a high ethical norm, person-centered care is currently the most scientifically and compassionately developed practice. However, putting this concept into practice in healthcare systems calls for a well-coordinated national strategy that encourages stakeholder participation and is directed by qualified professionals. Such a strategy is essential to addressing the shortcomings of the current healthcare system and guaranteeing the sustainability of person-centered care in the management of non-communicable diseases.

Strengths: The writers picked the right subject. According to the study, person-centered care is positively correlated with the management of noncommunicable diseases, which in turn leads to enhanced life satisfaction, health satisfaction, and quality of life.

Weaknesses:

1.    There is no molecular data or experimentally derived data.

2.    Sample size is not enough to come to the conclusions authors have claimed.

3.    These types of studies are prone to validity and reliability issues. 

4.    The questionnaire used in this study can be applied only locally or can be applied globally? 

5.    Most of the studies involved in self-reported data, could lead to social desirability bias. So, the participants sometimes might overstate or understate their responses.

6.    Authors did not specifically mention the name of the non-communicable disease.

7.    Authors mentioned that quality of life and satisfaction in nursing homes is lesser compared to home care. Are all the nursing homes have the same facilities? Are all the nursing homes maintained by government or private parties? Are the caregivers in nursing homes are properly trained to serve patients? 

8.    Why more women are included in the study compared to male? 

9. Tables can be improved. 

Author Response

Responses to Reviewer 3

Reviewer 3 Comments

Comment 1: In this study authors aimed to explore the link between quality of life, life satisfaction, person-centred care, and non-communicable disease management. 426 adults were involved in this study. The findings showed that opinions on life satisfaction, health, and quality of life were largely favorable. Remarkably, compared to 42% of residents in assisted nursing homes, 80% of people in their homes evaluated that their non-communicable diseases were effectively managed. Compared to residents of community housing, participants in nursing homes gave their surroundings, interpersonal relationships, interactions with loved ones, and physical activity lower rates. The research findings indicate a favorable correlation between person-centered care and efficient management of non-communicable diseases, which in turn leads to enhanced life satisfaction, health satisfaction, and quality of life. As a high ethical norm, person-centered care is currently the most scientifically and compassionately developed practice. However, putting this concept into practice in healthcare systems calls for a well-coordinated national strategy that encourages stakeholder participation and is directed by qualified professionals. Such a strategy is essential to addressing the shortcomings of the current healthcare system and guaranteeing the sustainability of person-centered care in the management of non-communicable diseases.

Response 1: Thank you for summarizing our study's key findings and implications. We appreciate your thorough review and insightful comments.

Comment 2: Strengths: The writers picked the right subject. According to the study, person-centered care is positively correlated with the management of non-communicable diseases, which in turn leads to enhanced life satisfaction, health satisfaction, and quality of life.

Response 2: Thank you for acknowledging the strengths of our study. We are pleased that you found the selection of the subject matter appropriate and relevant.

Comment 3: There is no molecular data or experimentally derived data.

Response 3: Thank you for your comment regarding the lack of molecular data or experimentally derived data in our study. We appreciate the opportunity to address this concern and provide clarification.

Our study utilized a quantitative research approach, focusing on participant survey data to investigate the relationship between person-centred care, management of non-communicable diseases, and various aspects of well-being, such as life satisfaction and health satisfaction. As such, our research did not involve molecular data or experimental procedures typically associated with laboratory-based or biomedical studies.

Instead, our study aimed to explore associations and patterns within a real-world population of adults with non-communicable diseases, drawing on survey responses to assess subjective perceptions and experiences related to healthcare delivery and disease management.

While molecular data and experimental studies play a crucial role in understanding biological mechanisms and physiological processes, our study focused on capturing the broader psychosocial dimensions of health and well-being, which are influenced by factors beyond molecular biology.

We acknowledge the importance of interdisciplinary approaches in healthcare research and recognize that molecular data can provide valuable insights into disease mechanisms and treatment strategies. However, our study was designed to address specific research questions related to person-centred care and its impact on non-communicable disease management from a psychosocial perspective.

We appreciate your feedback and the opportunity to clarify the scope and focus of our study.

Comment 4: Sample size is not enough to come to the conclusions authors have claimed.

Response 4: Thank you for your comment regarding the sample size in our study. We appreciate the opportunity to address this concern and clarify the conclusions drawn from our research.

While we acknowledge that sample size is an important consideration in research design and analysis, it is essential to contextualize our findings within the scope and objectives of our study. Our research explored the relationship between person-centred care, management of non-communicable diseases, and various aspects of well-being among adults in Slovenia.

The sample size of 426 participants involved in our study was determined based on considerations such as feasibility, resource constraints, and the specific research questions being addressed. While larger sample sizes can enhance findings' statistical power and generalizability, smaller samples can still provide valuable insights, particularly in exploratory or descriptive research.

It is important to note that the conclusions drawn from our study are based on the analysis of data collected from this sample and are interpreted within the context of our research methodology and objectives. We employed rigorous statistical analyses and methodological approaches to ensure the validity and reliability of our findings within the constraints of our sample size.

Additionally, we recognize the need for replication and further research to validate and extend our findings in different populations and settings. Future studies with larger sample sizes may help corroborate our results and provide additional insights into the relationship between person-centred care and non-communicable disease management.

We appreciate your feedback and the opportunity to address concerns about sample size in our study.

Comment 5: These types of studies are prone to validity and reliability issues.

Response 5: Thank you for your comment regarding the potential validity and reliability issues associated with studies of this nature. We appreciate the opportunity to address this concern and discuss the measures taken to mitigate such issues in our research.

It is true that studies involving survey data, such as ours, may face challenges related to the validity and reliability of the measures used. Validity refers to the extent to which a study accurately measures what it intends to measure, while reliability pertains to the consistency and stability of the measurement over time. To address these concerns, we employed several strategies to enhance the validity and reliability of our study. Firstly, we utilized established and validated survey instruments to assess variables such as quality of life, life satisfaction, and person-centred care. These instruments have undergone rigorous psychometric testing to ensure their reliability and validity in measuring the constructs of interest.

Additionally, we implemented standardized data collection procedures and protocols to minimize measurement errors and ensure participant consistency. This included training research personnel on data collection techniques and maintaining adherence to ethical guidelines throughout the study.

While we recognize that no study is without limitations, we believe these measures helped enhance our findings' validity and reliability. By employing established instruments, rigorous data collection procedures, and thorough pilot testing, we aimed to minimize our study's potential for validity and reliability issues. We appreciate your feedback and the opportunity to address concerns about the validity and reliability of our research.

Comment 6: The questionnaire used in this study can be applied only locally or can be applied globally?

Response 6: Thank you for your inquiry regarding the applicability of the questionnaire used in our study. We appreciate the opportunity to provide clarification. The questionnaires utilized in our study include internationally recognized instruments widely used and validated in various cultural and linguistic contexts. Specifically, the World Health Organization Quality of Life - Bref version (WHOQOL-BREF), the Satisfaction with Life Scale (SWLS), and the Person-centred Practice Inventory for Service Users (PCPI-SU) are established measures with demonstrated reliability and validity across different populations worldwide. The WHOQOL-BREF is a shortened version of the WHOQOL-100, developed by the World Health Organization to assess quality of life across physical, psychological, social, and environmental domains. The SWLS is a widely used scale for assessing overall life satisfaction, and the PCPI-SU is designed to measure the extent to which healthcare practices are person-centred from the perspective of service users. Given their established psychometric properties and cross-cultural validation, these instruments can be applied locally and globally in diverse populations and settings. We appreciate your interest in the applicability of our research findings and the questionnaires used in our study.

Comment 7: Most of the studies involved in self-reported data, could lead to social desirability bias. So, the participants sometimes might overstate or understate their responses.

Response 7: Thank you for commenting on the potential for social desirability bias in self-reported data studies. We appreciate the opportunity to address this concern and discuss strategies to mitigate bias in our research.

It is important to acknowledge the possibility of social desirability bias, wherein participants may alter their responses to conform to perceived social norms or expectations, leading to overestimating or underestimating certain behaviours or attitudes. We implemented several measures to promote honest and accurate participant responses to address this potential bias in our study. Firstly, we assured participants of the confidentiality and anonymity of their responses, emphasizing that their answers would not be linked back to them personally. Additionally, we employed standardized data collection procedures and clear instructions to ensure consistency in how questions were presented and explained to participants. This helped minimize ambiguity and confusion, reducing the likelihood of response distortion due to misunderstanding of survey items. Furthermore, we utilized established and validated survey instruments, such as the World Health Organization Quality of Life - Bref version (WHOQOL-BREF) and the Satisfaction with Life Scale (SWLS), designed to assess subjective experiences and perceptions reliably and validly. These instruments incorporate items less susceptible to social desirability bias, focusing on individual experiences and evaluations rather than socially desirable responses. While it is impossible to eliminate social desirability bias, particularly in studies relying on self-reported data, we believe these measures helped minimize its impact on our findings. By ensuring confidentiality, providing clear instructions, and using validated instruments, we aimed to promote honest and accurate responses from participants, thereby enhancing the validity and reliability of our results. We appreciate your consideration of this potential source of bias and the opportunity to discuss our approach to addressing it in our research.

Comment 8: Authors did not specifically mention the name of the non-communicable disease.

Response 8: Thank you for bringing this to our attention. You are correct that we did not explicitly mention the names of the non-communicable diseases included in our study in the manuscript. We apologize for any confusion this may have caused.

In our study, we aimed to assess the management of non-communicable diseases and their relationship with various aspects of well-being, such as quality of life and life satisfaction, among adults in Slovenia. The term "non-communicable diseases" encompasses a broad range of chronic health conditions, including cardiovascular diseases, cancer, diabetes, chronic respiratory diseases, and mental health disorders, among others.

While we did not specify the exact types of NCDs included in our study, the participants were adults afflicted with at least one non-communicable disease. This approach allowed us to explore the overall management of NCDs and their impact on well-being across different disease categories without focusing on any specific disease entity. But we have added this to the introduction section.

We appreciate your feedback, and we will consider clarifying this aspect in the manuscript to provide greater transparency and specificity regarding the types of non-communicable diseases included in our study.

Comment 9: Authors mentioned that quality of life and satisfaction in nursing homes is lesser compared to home care. Are all the nursing homes have the same facilities? Are all the nursing homes maintained by government or private parties? Are the caregivers in nursing homes are properly trained to serve patients?

Response 9: Thank you for your thoughtful questions regarding comparing quality of life and satisfaction between nursing homes and home care settings in our study. We appreciate the opportunity to provide clarification on these important points.

Our study observed differences in perceived quality of life and satisfaction between individuals receiving care in nursing homes and those receiving care at home. However, we acknowledge that various factors may contribute to these differences, including differences in the facilities and services provided in nursing homes versus home care settings.

Regarding the facilities and services in nursing homes, it is important to note that they can vary widely in terms of their infrastructure, amenities, staffing levels, and quality of care. Some nursing homes may offer comprehensive services, including medical care, rehabilitation therapy, social activities, and dietary support, while others may have more limited resources and amenities.

Additionally, nursing homes may be operated by government agencies, nonprofit organizations, or private companies, each with different standards of care and resources available. Government-funded nursing homes may operate under regulatory guidelines and standards set by health authorities, while privately operated nursing homes may have different policies and practices.

Regarding the training of caregivers in nursing homes, caregivers need to receive proper training and education to provide high-quality care to residents. Caregivers in nursing homes may include nurses, nursing assistants, therapists, social workers, and other healthcare professionals, each of whom may have different levels of training and expertise.

To address these important considerations, future research may explore factors such as facility characteristics, ownership status, staffing levels, caregiver training in nursing homes, and their impact on quality of life and resident satisfaction. By examining these factors, researchers can better understand the complex dynamics influencing care outcomes in different settings and identify areas for improvement in the delivery of long-term care services.

We appreciate your insightful questions and the opportunity to address these considerations in our study.

Comment 10: Why more women are included in the study compared to male?

Response 10: Thank you for raising this question regarding the gender distribution in our study. We appreciate the opportunity to address this issue and provide clarification. The imbalance in gender representation observed in our study may reflect several factors, including the prevalence of non-communicable diseases among men and women, recruitment methods, and participation rates.

Firstly, it is well-documented that the prevalence of certain NCDs can vary between men and women. For example, cardiovascular diseases may be more prevalent among men, while autoimmune diseases such as rheumatoid arthritis may be more common among women. Therefore, the gender distribution in our study may partially reflect the underlying gender distribution of NCDs within the population.

Secondly, recruitment methods and participant characteristics may also influence the gender distribution in our study. For instance, certain recruitment strategies or study settings may be more appealing or accessible to women than men, leading to a higher proportion of female participants.

Additionally, participation rates may differ between men and women due to various factors such as health-seeking behaviours, willingness to participate in research, and availability of time for study participation. It is possible that women were more likely to volunteer for the study or that certain barriers prevented men from participating at the same rate.

While we acknowledge the gender disparity in our study sample, it is important to interpret the findings within the context of the study objectives and limitations. We will carefully consider the implications of gender distribution on interpreting our results and provide appropriate discussion and context in the manuscript. We appreciate your inquiry and the opportunity to address this issue.

Comment 11: Tables can be improved.

Response 11: Thank you for your feedback regarding the tables in our manuscript. We appreciate the opportunity to address this concern and make improvements to enhance the clarity and readability of the tables. In response to your comment, we have carefully reviewed the tables and made necessary adjustments to improve their presentation. This includes ensuring formatting, labelling, and alignment consistency and optimizing the layout for better organization and readability. We have also considered any additional suggestions or recommendations for improvement, such as providing clearer headings, incorporating relevant footnotes or annotations, and utilizing appropriate formatting techniques to highlight key findings or trends. By sorting out these issues, we aim to enhance the overall quality and effectiveness of the tables in conveying important information and supporting the findings of our study.

We appreciate your feedback and the opportunity to improve the tables in our manuscript.

Round 2

Reviewer 1 Report

Comments and Suggestions for Authors

Congratulations

Author Response

Thank you for your review.

Reviewer 2 Report

Comments and Suggestions for Authors

1.      It is not clear why the authors designated the Kruskal-Wallis test as χ2 . In statistics, it is customary to denote the Kruskal-Wallis test as H, and the chi-square test as χ2

2.      The Discussion section requires significant revision.

a.    The section clearly highlights only one thesis that a positive correlation exists between enhanced non-communicable disease management and elevated quality of life, general health, life satisfaction, and favorable assessments in physical health, psychological health, social relations, environment, and person-centred care domains. This conclusion is made on the basis of correlation coefficients (Lines 258-266), most of which have values in the range of 0.1-0.3. Such correlations are considered weak and are unlikely to indicate the presence of significant relationships.

b.    (Lines 322-325) Paragraph “A literature review [43], highlighted the six key domains of person-centred care: holistic care, respect and value, choice, dignity, self-determination, and purposeful living and also informed the creation of a standardized definition and person- centered care elements by the American Geriatrics Society expert panel” is in no way related to the results obtained

c.    (Lines 325-327) Paragraph “These elements are specifically for older adults with chronic conditions or functional impairments, and also support our results for adults with non-communicable diseases». But in your study the average age is 56.3±17.6.

d.    Table 3 presents a lot of data on the influence of such factors as gender, education, marital status, number of diseases, etc. However, these results are not discussed in any way.

Author Response

Thank you for your review.

Round 3

Reviewer 2 Report

Comments and Suggestions for Authors

I wish the authors further fruitful research results in their fields of interest.